# Assessing the value of complex refractive index and particle density for calibration of low-cost particle matter sensor for size-resolved particle count and PM2.5 measurements

**Ching-Hsuan Huang**[1], **Jiayang He**[2], **Elena Austin**[1], **Edmund Seto**[1], **Igor Novosselov**[2]*

**1** Department of Environmental and Occupational Health Sciences, School of Public Health, University of Washington, Seattle, Washington, United States of America, **2** Department of Mechanical Engineering, College of Engineering, University of Washington, Seattle, Washington, United States of America

* ivn@uw.edu

**Data Availability Statement:** All relevant data are within the manuscript and its Supporting Information files.

## Abstract

Low-cost optical scattering particulate matter (PM) sensors report total or size-specific particle counts and mass concentrations. The PM concentration and size are estimated by the original equipment manufacturer (OEM) proprietary algorithms, which have inherent limitations since particle scattering depends on particles' properties such as size, shape, and complex index of refraction (CRI) as well as environmental parameters such as temperature and relative humidity (RH). As low-cost PM sensors are not able to resolve individual particles, there is a need to characterize and calibrate sensors' performance under a controlled environment. Here, we present improved calibration algorithms for Plantower PMS A003 sensor for mass indices and size-resolved number concentration. An aerosol chamber experimental protocol was used to evaluate sensor-to-sensor data reproducibility. The calibration was performed using four polydisperse test aerosols. The particle size distribution OEM calibration for PMS A003 sensor did not agree with the reference single particle sizer measurements. For the number concentration calibration, the linear model without adjusting for the aerosol properties and environmental conditions yields an absolute error (NMAE) of ~ 4.0% compared to the reference instrument. The calibration models adjusted for particle CRI and density account for non-linearity in the OEM's mass concentrations estimates with NMAE within 5.0%. The calibration algorithms developed in this study can be used in indoor air quality monitoring, occupational/industrial exposure assessments, or near-source monitoring scenarios where field calibration might be challenging.

## Introduction

The direct measurement of time- and size-resolved particle matter (PM) concentrations is essential to health-related applications, such as exposure assessments and air quality (AQ) studies, but are challenging to implement at fine spatial and temporal scales. Human exposure

**Funding:** This work was partially supported by the National Institute of Environmental Health Sciences [grant numbers 1R21ES024715 and 1R33ES024715]; and the National Institute of Biomedical Imaging and Bioengineering [grant number U01 EB021923].

**Competing interests:** The authors have declared that no competing interests exist.

to PM is associated with multiple adverse health effects, including cardiovascular disease, cardiopulmonary disease, and lung cancer [1–7]. Estimates show that approximately 3% of cardiopulmonary and 5% of lung cancer deaths are attributed to exposures to $PM_{2.5}$ (particles less than 2.5 μm in diameter) globally [8]. Particle deposition in the human respiratory tract and the resultant adverse health effects depend on the particles' size distribution [9, 10]. PM concentration varies significantly in space and time across community settings [11, 12]. Hence, time- and size-resolved PM measurements are more informative than traditional total PM weight measurements for assessing adverse health effects. As part of the Clean Air Act, the National Ambient Air Quality Standard (NAAQS) set by the U.S. Environmental Protection Agency (EPA) has adopted and established monitoring requirements for six criteria air pollutants, including $PM_{2.5}$ and $PM_{10}$ [13, 14]. However, the sparse spatial distribution of government monitoring sites makes fine spatial scale exposure assessment challenging [8]. Traditional PM measurement instruments are large and expensive, thus have limited use in high spatial and temporal resolution mapping applications; these applications instead demand compact, low-cost sensors with reliable performance.

Low-cost PM sensors find increasing use in various applications, including monitoring AQ in the outdoor [15–18] and indoor environment [19–21] by academic researchers and citizen scientists. The low-cost sensor networks have the potential to provide high spatial and temporal and resolution, identifying pollution sources and hotspots, which in turn can lead to the development of intervention strategies for exposure assessment and intervention strategies for susceptible individuals. Time-resolved exposure data from wearable monitors can be used to assess individual exposure in near real-time [22].

As low-cost sensors find applications in pollution monitoring, and there is a need to characterize and calibrate their performance under various conditions, calibration in controlled environments with standardized test aerosols can provide the basis for such assessments. Various studies have evaluated the performance of low-cost PM sensors in laboratory and field settings [23–30]; these reports show that low-cost sensors yield usable data when calibrated against research-grade reference instruments, although some drawbacks have been reported. One common concern is that calibrations for number concentrations have not been reported, and the mass concentration of the low-cost PM sensors is based on numerous assumptions. Second, there is a lack of information on low-cost sensors' ability to assess particle size distributions, which is critical for assessing health and environmental impacts. Third, calibrations based on short-term field colocations with reference instruments are often limited by the range of particle properties, concentrations, and environmental conditions and thus cannot be generalized to other studies. This is a concern because with improving air quality in the developed nations, the typical $PM_{2.5}$ levels are relatively low (<20 μg/m$^3$); however, PM concentration during wildfires [31] and in occupational settings [32, 33] often exceeds regulatory limits for short periods. In developing countries with less strict regulations, the PM level associated with, e.g., traffic emissions [34], agricultural waste burning [35], indoor cooking [36] is significantly higher. In these settings, field colocations with reference instruments required for calibration studies can be challenging. Thus, evaluating low-cost PM sensors' performance under high and low loading conditions is necessary if the sensors were to be used in epidemiological studies and PM surveillance networks.

Low-cost optical PM sensors rely on elastic light scattering to measure time- and size-resolved PM concentrations; they are widely used in aerosol research, particularly when measuring particles in the 0.5 μm to 10 μm range. Aerosol photometers that measure the bulk light scatter of multiple particles simultaneously have limited success in measuring mass concentration [30]. Typical low-cost (<$100) particle monitors often yield unreliable number concentrations data [37], but PM mass estimation error can be as high as 1000% [38]. Also, low-cost

sensor measurements may suffer from sensor-to-sensor variability due to a lack of quality control and differences between individual components [30, 37]. Sensor geometry can be optimized to reduce the effect of particle CRI. Researchers have addressed CRI sensitivity by designing optical particle sizers (OPSs) that measure scattered light at multiple different angles simultaneously [39] or by employing dual-wavelength techniques [40]. However, these solutions involve complex and expensive components not suitable for compact, low-cost devices. Optimizing the detector angle relative to the excitation beam can reduce dependency on CRI [41]; however, this approach has not been translated to high volume production.

Some commercially available low-cost sensors provide output in total particle counts or particle mass concentrations, and some provide size-specific counts or mass concentrations. These quantities are not measured directly as an individual particle's scattering signature (as in the single particle counters) but are estimated by the OEM proprietary algorithms. These algorithms have inherent limitations because particle scattering depends on the particles' composition, size, shape, and CRI [42]. A common workaround is to collect PM on a filter after or in parallel with the OPS measurements. The filters are analyzed to determine their average particle optical properties; these data are then used to correct the optical measurements after the fact.

Environmental conditions can affect sensor output, e.g., a non-linear response has been reported with increasing RH [43–47]. High humidity (RH > 75%) creates challenges for particle instruments; e.g., significant variations were observed between different commercially available instruments, such as Nova PM sensor [43] and personal DataRAM [45]. In addition, the RH measurement approach could also affect the sensor output [43, 44], e.g., the RH measurement based on reference monitoring site rather than inside the sensor enclosure may be different due to the microenvironment and transient effects. The selection of reference instruments with different measuring principles may also influence the calibration of low-cost sensors. For example, the calibration of the Plantower PM sensor in Jayaratne et al., 2018 was based on the tapered element oscillating microbalance (TEOM), while Zusman et al., 2020 calibrated the same sensor against the beta attenuation monitor (BAM) and federal reference method (FRM) measurements [29, 44]. The integrated mass measurements cannot account for temporal particle size and concentration variation during the calibration experiment. The instruments that directly measure aerosol size and concentration can be a better fit for sensor calibration [30, 48]. The calibration against aerodynamic particle sizer (APS) or single optical particle counter instruments can potentially provide a more robust calibration for low-cost optical particle sensors. Correlating particle diameter measured low-cost sensor to aerodynamic diameter measured by an APS is relevant since the aerodynamic diameter determines particle deposition in the respiratory tract.

This study presents calibration for PMS A003 (Beijing Plantower Co., Ltd, China; referred to as PMS hereafter) sensors as a function of particle sizes and concentration, as well as the $PM_{2.5}$ and $PM_1$ indices. The calibration is based on four polydisperse standard testing aerosols, including the Arizona Test Dust (ATD), two types of ceramic particles, and NaCl particles. The PMS data from six sensors were calibrated against the APS for particle size range 0.5–10 μm and number concentration in the range of 0–1000 #/cc. A standardized laboratory experimental protocol was developed to control the PM concentration, environmental conditions and to assess sensor-to-sensor reproducibility.

## Materials and methods

### Plantower PMS A003 and sensor test platform

The low-cost sensor PMS A003 was evaluated. The sensor's photodiode is positioned perpendicular to the excitation beam and measures the ensemble scattering of particles in the optical

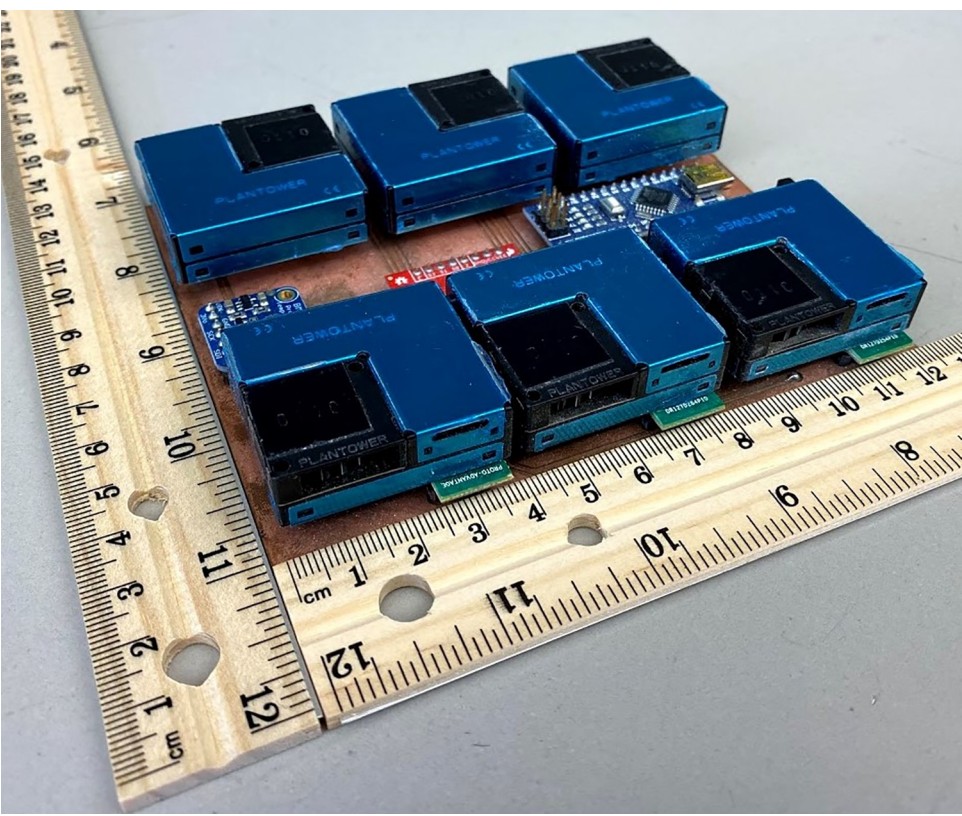

**Fig 1. Photograph test platform, consisting of six PMS units mounted on the PCB, a temperature and humidity sensor, a multiplexer, and an Arduino microcontroller.**

volume. The scattering light intensity is then converted to a voltage signal to estimate PM number density and mass concentration using a proprietary calibration algorithm. The PMS provides estimated particle counts in six size bins with the optical diameter in 0.3–10 μm (#/0.1L) range and mass concentration (μg/m$^3$) for PM$_1$, PM$_{2.5,}$ and PM$_{10}$. The mass concentrations are reported for two settings: "factory" and "atmospheric" conditions. The factory condition applies a correction factor (CF) of unity to the concentration measured, whereas the "atmospheric" condition is designed for use in the ambient environment.

Six PMS units were installed on a custom printed circuit board (PCB), which also included a Bosch BME680 temperature and relative humidity (RH) sensor (Fig 1). All sensors were connected to an Arduino Nano microcontroller through a data selector (multiplexer NXP 74HC4051 breakout board, Sparkfun, Boulder, CO). The controller collects data from the six PMS sensors and an RH and temperature sensor simultaneously with the data acquisition rate of 1 Hz. The data reported in "factory" mode were used in the analysis.

The reference instrument used in this study is the TSI Aerodynamic Particle Sizer (APS) 3321 spectrometer. APS measures both the aerodynamic size and optical size of a particle. Using the time-of-flight principle, the APS measures size-resolved particle counts with aerodynamic particle diameter (AD) of 0.523 to 20 μm in 52 size bins. The lower detection limit for optical size is 0.37 μm. The APS's optical sensor detects particles with AD<0.523 μm, but they can not be resolved based on their aerodynamic size. Thus these particles are assigned to a single bin <0.523 μm. The aerodynamic size determined the particle's aerodynamic behavior, such as settling velocity or penetration into the respiratory tract. Thus, we evaluate the

correlation between the PMS number concentrations and the APS aerodynamic size bin number concentrations. The instrument estimates mass concentration by assuming spherical particles and particle density input. APS reports particle concentrations with a 5-second resolution. In the experiments, the sampling inlet was placed near the PMS sensors. Per APS specifications, the maximum recommended particle concentration is 1000 #/cm$^3$ at 0.5 μm with $< 5\%$ coincidence. Therefore, the total number concentration of the aerosols in the test chamber was maintained below 1000 #/cc ($10^5$ #/0.1L).

## Aerosol chamber tests

We tested four polydisperse aerosols: Arizona Test Dust (ATD) (Powder Technology Incorporated, Arden Hills, MN), polydisperse W210, and W410 ceramic particles (3M™, St. Paul, MN), and sodium chloride (NaCl) particles. NaCl particles were generated by nebulizing the aqueous solution of 10% wt [49]. The challenge aerosols' properties and typical size distributions are summarized in Table 1 and S3 Fig., respectively. The experiments were conducted in a custom-built aerosol chamber (0.56 m × 0.52 m × 0.42 m) (Fig 2). A full description of the chamber can be found in ref [50]. The PMS sensor platform was placed in the well-mixed aerosol test chamber, elevated to the same height as the APS inlet. The APS sampled particle-laden air through static-dissipative tubing to eliminate electrostatic losses in the tubing. Particles were generated using a medical nebulizer (MADA Up-Mist Medication Nebulizer) [51]. During the experiments, the RH was controlled by nebulizing deionized water using a separate nebulizer or introducing dry filtered air; tests were conducted in the range of RH = 17%– 80%. Two mixing fans inside the chamber provided well-mixed conditions through the experiments; particle concentration was continuously monitored.

We controlled the aerosol generation rate by adjusting the compressed air flow rate to the nebulizer. The aerosol generation was stopped when the total number concentration (based on the APS count) reached 1000 #/cm$^3$. Then, the particle concentration was allowed to decay as the chamber was evacuated at a rate of 9.8 L/min; the make-up air entering the test was aspirated through a HEPA filter. The sensor array data and the APS data were acquired via two universal serial buses (USB) cables in real-time until the total number concentration from the APS reached 15 #/cm$^3$.

## Data analysis and modeling

The collected data with concentrations $> 1000$ #/ cm$^3$ were removed. The number concentration reported by the APS was aggregated as summarized in Table 2 to match the cumulative number concentrations of the PMS. The 1-second PMS measurement and 5-second APS measurement were aggregated to obtain 1-minute averaged data for calibration. The smallest size bin of the APS ($< 0.523$ μm) was used as a reference for calibrating PMS size bin $> 0.3$ μm.

Table 1. Characteristics of the standard testing aerosols used in the study [52].

| Aerosol | ATD | W210 | W410 | NaCl |
|---|---|---|---|---|
| Composition | SiO$_2$, Al$_2$O$_3$, Fe$_2$O$_3$, Na$_2$O [a] | Alkali aluminosilicate ceramic | Alkali aluminosilicate ceramic | Sodium Chloride |
| Assumed density (g/cm$^3$) | 2.5–2.7 [b] | 2.4 | 2.4 | 1.03 |
| CRI | 1.63 | 1.53 | 1.53 | 1.54 |

[a] Four major components are listed.

[b] For analysis purposes, a density of 2.6 g/cm$^3$ was used.

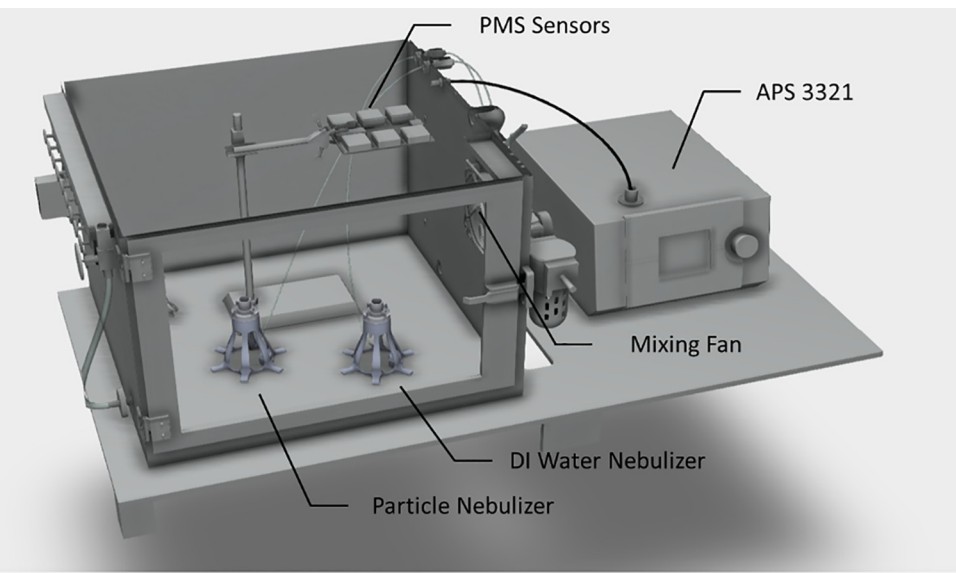

**Fig 2. A 3D view of the experimental setup.**

Fig 3 shows the data from all six PMS sensors during the typical experimental run. In all experiments, the time-series of the uncalibrated concentration measurements from the six PMS sensors were consistent for all size bins (Pearson correlation coefficient (r) > 0.98) (S4–S12 Figs). The data consistency between the six sensors allows us to develop generalized models by fusing the readings from all sensors and then correlating the data against the APS measurement with matching time stamps for each size bin. The calibrations models of the following form were fit for number concentration data from the APS and PMS:

$$APS_t = \beta_0 + \beta_1 \, PMS_t + \varepsilon_t \tag{1}$$

where $APS_t$ is the number concentration for each aggregated APS size bin at timestamp t; $PMS_t$ is the linear term of the PMS measurement (the number concentration of each PMS size bin) at timestamp t; $RH_t$ is the RH measurement of the Bosch BME680 sensor at timestamp t; $\beta_0$ and $\beta_1$ are regression coefficients; $\varepsilon_t$ is the residual. In addition to Eq (1), other forms of

**Table 2. The PMS manufacturer's specified size bins and mass indices with the corresponding reference APS aerodynamic size bins for calibration.**

| PMS indices | Reference APS indices |
|---|---|
| *Number concentration* | |
| > 0.3 μm | counts aggregated from all size bins (< 0.523 μm and 0.542–19.81 μm) |
| > 0.5 μm | counts aggregated from size bins 0.542–19.81 μm |
| > 1 μm | counts aggregated from size bins 1.037–19.81 μm |
| > 2.5 μm | counts aggregated from size bins 2.642–19.81 μm |
| > 5 μm | counts aggregated from size bins 5.048–19.81 μm |
| > 10 μm | counts aggregated from size bins 10.37–19.81 μm |
| *Mass concentration* | |
| $PM_1$ | mass aggregated from size bin < 0.523 μm– 0.965 μm |
| $PM_{2.5}$ | mass aggregated from size bin < 0.523 μm– 2.458 μm |
| $PM_{10}$ | mass aggregated from size bin < 0.523 μm– 9.647 μm |

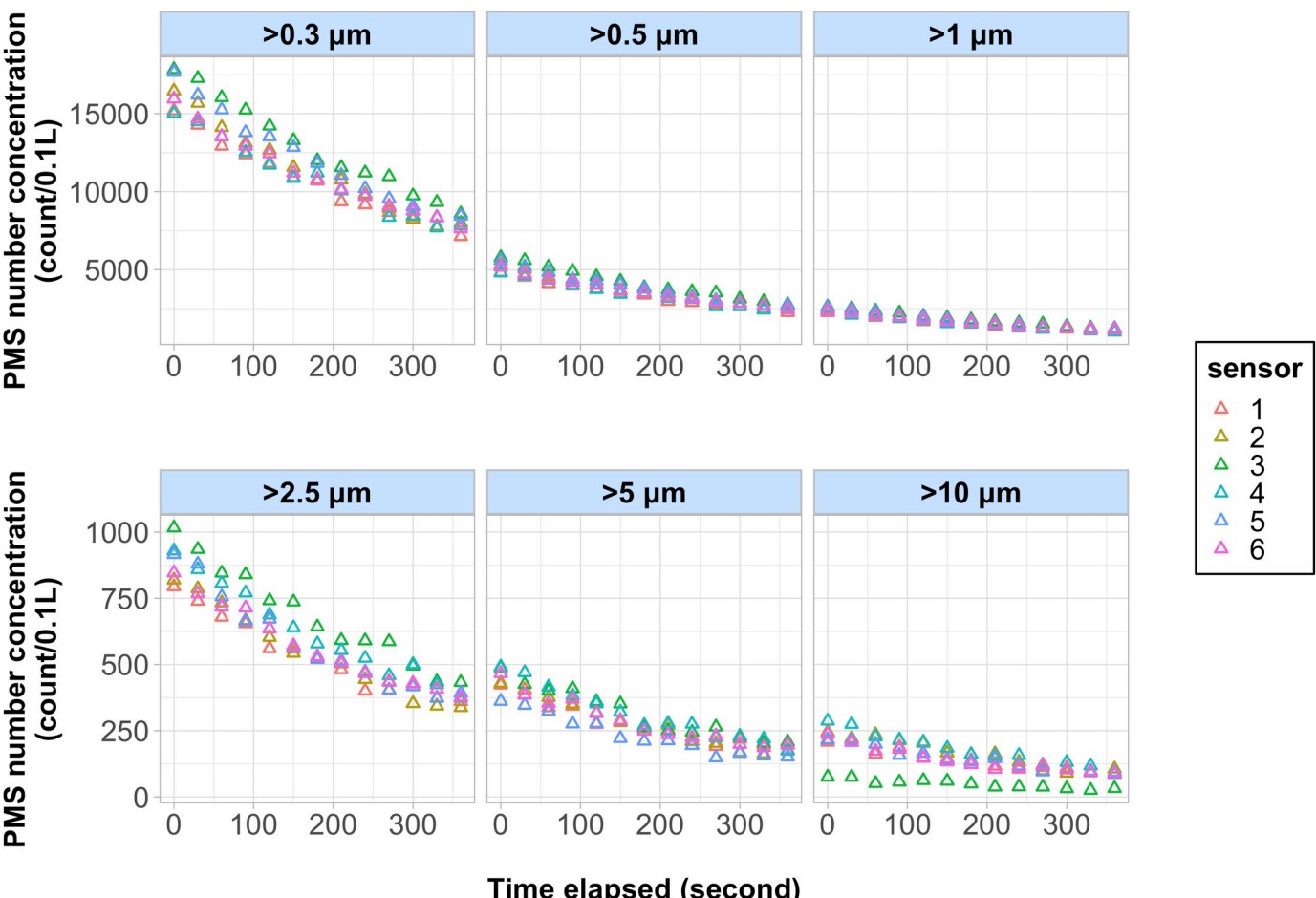

**Fig 3. Time-series plots of the uncalibrated, 1-second number concentration measurement from the six PMS sensors are presented.** The experiments were conducted under 30% relative humidity with W210 aerosols.

linear models adjusted for relative humidity, particle density, and CRI were evaluated (S1 Table). For calibration of mass concentration, models including quadratic terms of the PMS measurement were evaluated (S1 Table). Since the test temperature variations were within ±2°C, the temperature was not included as a variable in the models. Based on *a priori* assumption that the PMS particle count and mass indices should be zero when the APS count is zero, the intercept ($\beta_0$) of the models was set to zero. The number of terms included in the optimal calibration model for each size bin was determined based on the Bayesian Information Criterion (BIC). The models with lower BIC were chosen as the optimal models. After identifying the optimal models using BIC and estimating model coefficients, the model was then applied to the pre-calibrated 1-minute PMS measurement to produce the post-calibrated concentrations for model evaluation. Calibration performance was assessed using the normalized mean absolute error (NMAE), which was calculated using the following equation [53]:

$$\text{NMAE } (\%) = \frac{\text{Mean}(|C_{\text{PMS\_post-cal}} - C_{\text{APS}}|)}{\text{Mean}(C_{\text{APS}})} \quad (2)$$

where $C_{\text{PMS\_post-cal}}$ is the post-calibrated 1-minute averaged PMS concentration and $C_{\text{APS}}$ is

the 1-minute averaged APS concentration. The linear models were fitted using the *lm* function in R. All the analyses were conducted using R version 3.6.3.

## Results and discussion

### Test conditions

During the experiments, the average temperature in the chamber was 24.8˚C (range: 23.2 to 26.6˚C), the RH was varied in the range of 17.5–79.4%, all experiments were performed at atmospheric pressure conditions. The one-minute APS total number concentration averaged 237.9 #/cm$^3$ (range: 8.5 to 985.9 #/cm$^3$). The one-minute PM$_{2.5}$ measurement from the APS and PMS before calibration (6 sensors pooled together) averaged 106.0 μg/m$^3$ (range: 1.9 to 641.3 μg/m$^3$) and 51.5 μg/m$^3$ (range: 0 to 218.8 μg/m$^3$), respectively.

### Particle size distribution

The particle size distribution of each test aerosol by the APS is shown in S3 Fig. The NaCl particles (from the nebulized liquid solution) were the smallest among the test aerosols, with nearly all particles < 3 μm. The W410 mixture had slightly larger particles than W210 and had the same CRI as W210 [52]. Fig 4 shows the typical particle size distributions reported by the PMS and the APS; the APS bins were aggregated to match the PMS. For all aerosols and all tested concentrations, the PMS appeared to underestimate particle counts for the size bin 0.5–1 μm and 1–2.5 μm. For larger size bins (2.5–5 μm and 5–10 μm), the PMS overestimate the particle counts. The particle count varies significantly in the lowest size bin (PMS: d$_p$ = 0.3–0.5 μm; APS: d$_p$ <0.523 μm). Both the APS and PMS use the optical channel. Since APS is a single particle instrument, its detection limit is based on the excitation wavelength, photodetector sensitivity, and particle optical properties; it is reported to be 0.37 μm. The PMS does not count every single particle; it relies on the internal calibration of the bulk scattering signal.

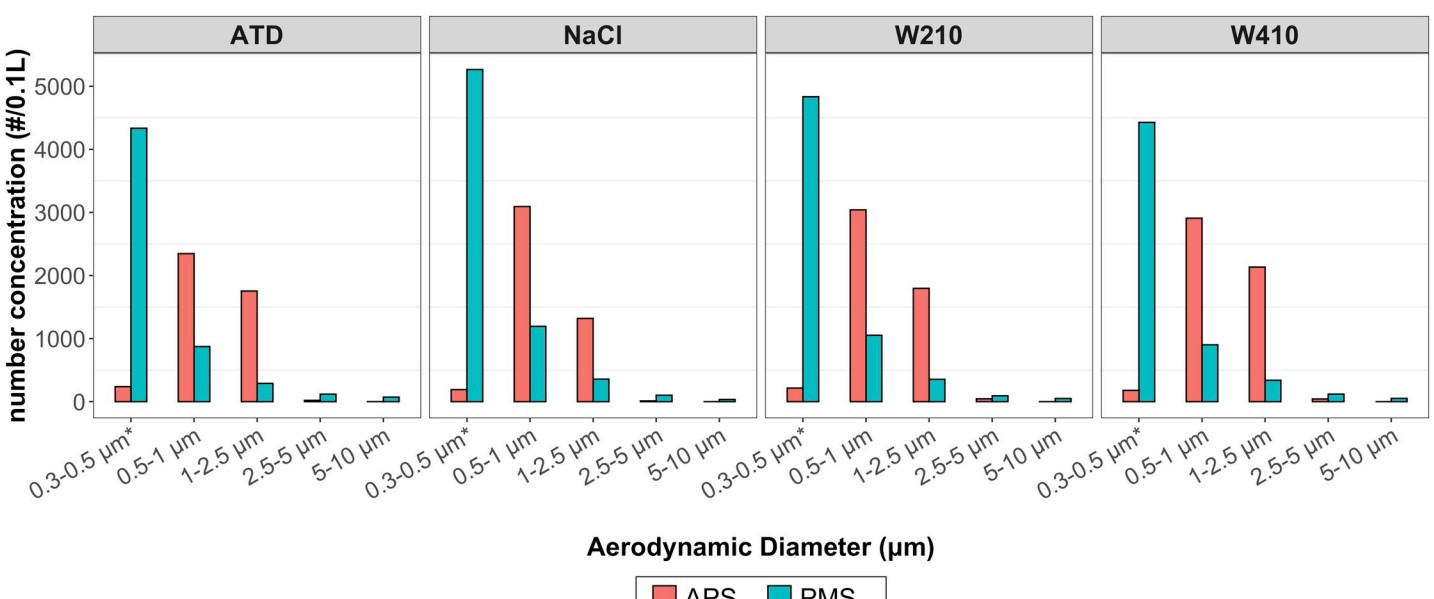

**Fig 4. A comparison of the size distribution measured by the APS and PMS using OEM calibration: Data from one of the six PMS for ATD, NaCl, W210, and W410 particles taken 15 minutes after the aerosols were introduced into the chamber.** The data from APS bins are aggregated to match the size bins reported by PMS. For PMS bin 0.3–0.5 μm, APS bin < 0.523 μm was used for comparison.

The PMS OEM calibration significantly overestimates the counts in the 0.3–0.5 μm size range. For PMS, we measured the lower detection limit for the number concentration for the NaCl particles. In NaCl particle experiments, the initial (non-zero) response in the most sensitive PMS bin ($d_p > 0.3$ μm) was observed at ~ 83 #/0.1 L as measured by APS. The PMS sensors seemed to follow the overall trends in the size distribution of the particles greater than 0.5 μm; however, the number concentration and particle sizing do not agree with the single-particle counter. Thus, calibration is needed if PMS is to be used for PM number concentration measurement.

## Model fit

Despite the apparent shift in size distribution shown by Fig 4, a matrix of Pearson correlations between the PMS number concentrations (before calibration) and the APS reference number concentrations for different size ranges suggests that the matching size bins between the two instruments are well-correlated (S1 Fig). Notably, the PMS number concentration data correlated well with the APS for size bin up to 2.5 μm ($r > 0.97$). For measurement of size bin larger than 2.5 μm, the PMS exhibited moderate correlation with the APS ($r < 0.78$). The worst correlation was observed for particles > 5 μm.

A similar Pearson correlation matrix comparing the mass concentrations measured by the PMS (before calibration) and the APS for different particle size ranges suggests a good correlation between matching sizes (S2 Fig). The $PM_1$, $PM_{2.5}$, and $PM_{10}$ measurements by the PMS all exhibited high correlations with their corresponding sizes measured by the APS ($r > 0.90$).

Because of the close correlations between corresponding APS and PMS size-specific measurements (S1 and S2 Figs), the sizes listed in Table 2 were chosen to develop calibration models for both PM number concentration and PM mass concentration. For example, APS size bin > 0.5 μm number concentrations was chosen as the reference (independent variable) for calibrating the PMS size bin > 0.5 μm number concentrations (dependent variable). Based on the same rationale, the corresponding APS mass concentration measurement was chosen as the reference for calibrating the PMS mass concentration index. The densities for each testing aerosol are shown in Table 1; these were used to determine the APS mass concentration measurement for calibration.

After fitting a set of alternative calibration model forms to the APS and PMS number concentration data, the results show excellent $R^2$ and low NMAE for >0.3 μm, >0.5 μm, and >1 μm size bins when the full range of concentrations from 0–1000 #/cm³ was used for fitting (Table 3 and S2 Table). However, the model performance was worse for larger size bins, i.e., >2.5 μm, >5 μm, and >10 μm size bins. Based on the previous findings on the impacts of relative humidity on optical particle sensor output and particle optical properties, the models adjusted for CRI and RH were considered in addition to the linear model, see Table 3. For most size ranges, the relatively simple linear model without the CRI dependent term performed nearly as well as the models with the additional parameters. The BIC suggests that the models (shown in bold in S2 Table) with the adjustment for CRI and RH have similar NMAE. The table also includes the models based on the lower concentration data (data points with APS total number concentration <100 #/cm³). The models based on the low PM concentration do not perform as well as the full range concentration models.

The OEM calibration for PM mass concentration showed significantly better agreement with the reference instrument than the OEM number density calibration. An additional quadratic term was included for fitting mass concentration data, as shown in Table 4 and S3 Table. Similar to the number concentration models, restricting the mass concentration model to only lower concentrations (data points with APS total number concentration <100 #/cm³)

**Table 3. Summary of the calibration models for number concentration, $R^2$, BIC, and the NMAE of the calibration models.**

| Indices | Equation | Regression [a] | $R^2$ | BIC | NMAE |
|---------|----------|----------------|-------|-----|------|
| *Full concentration range (APS total number concentration 0–1000 #/ cm³) (n = 4,134)* | | | | | |
| >0.3 μm | Linear | y = 5.93 x | 0.99 | 78723 | 2.20% |
| | Linear + CRI + RH | y = 6.00 x—1090 CRI + 28.23 RH | 0.99 | 78567 | 2.06% |
| >0.5 μm | Linear | y = 14.17 x | 0.98 | 79716 | 2.92% |
| | Linear + CRI + RH | y = 14.40 x—1434 CRI + 40.68 RH | 0.98 | 79518 | 2.78% |
| >1 μm | Linear | y = 14.85 x | 0.96 | 76002 | 2.88% |
| | Linear + CRI + RH | y = 14.98 x—784.93 CRI + 26.40 RH | 0.97 | 75884 | 2.89% |
| >2.5 μm | Linear | y = 2.20 x | 0.66 | 62906 | 3.87% |
| | Linear + CRI + RH | y = 2.42 x—156.71 CRI + 3.38 RH | 0.68 | 62695 | 3.95% |
| >5 μm | Linear | y = 0.11 x | 0.31 | 38958 | 2.71% |
| | Linear + CRI + RH | y = 0.14 x—2.41 CRI—0.06 RH | 0.33 | 38848 | 2.83% |
| >10 μm | Linear | y = 0.11 x | 0.70 | 8117 | 3.66% |
| | Linear + CRI + RH | y = 0.14 x—2.41 CRI—0.06 RH | 0.71 | 8000 | 3.68% |
| *Lower concentration range (APS total number concentration < 100 #/ cm³) (n = 1,838)* | | | | | |
| >0.3 μm | Linear | y = 4.84 x | 0.97 | 30263 | 7.96% |
| | Linear + CRI + RH | y = 4.94 x—358.12 CRI + 13.68 RH | 0.97 | 30235 | 7.94% |
| >0.5 μm | Linear | y = 14.17 x | 0.93 | 30285 | 10.39% |
| | Linear + CRI + RH | y = 10.83 x—385.61 CRI + 18.91 RH | 0.94 | 30234 | 10.23% |
| >1 μm | Linear | y = 14.85 x | 0.92 | 26963 | 8.50% |
| | Linear + CRI + RH | y = 11.10 x—105.04 CRI + 10.59 RH | 0.93 | 26713 | 7.57% |
| >2.5 μm | Linear | y = 2.20 x | 0.60 | 14298 | 7.73% |
| | Linear + CRI + RH | y = 0.45 x—8.38 CRI + 0.48 RH | 0.66 | 14018 | 7.22% |
| >5 μm | Linear | y = 0.11 x | 0.35 | -2612 | 11.52% |
| | Linear + CRI + RH | y = 0.003 x + 0.007 CRI + 0.002 RH | 0.44 | -2869 | 11.91% |
| >10 μm | Linear | y = 0.11 x | 0.19 | -5846 | 15.16% |
| | Linear + CRI + RH | y = 0.002 x + 0.003 CRI + 0.0003 RH | 0.22 | -5920 | 17.91% |

[a] y: APS measurement; x: PMS measurement.

Definition of abbreviations: n = number of data points.

resulted in worse performance vs. model based on the entire concentration range. The optimal models (shown in bold in S3 Table) included quadratic terms of PMS measurement, terms related to particle properties, and environmental conditions (CRI, density, and in some cases, RH) resulted in NMAE < 3.4% for the entire concentration range. Compared to the relatively simple linear models without these added parameters (CRI, density, and RH terms), the improvements in NMAE for models adjusted for these parameters tended to be larger than those observed for the number concentration models. The inclusion of quadratic term of PMS measurement in these optimal models also highlighted the non-linearity in the OEM's mass concentrations estimates (Table 4 and S3 Table). CRI adjustment did not produce a significantly better fit for mass concentration calibration. In addition, the $R^2$ values of the $PM_{10}$ models (ranged between 0.85–0.88) are lower than the $R^2$ values of the $PM_{2.5}$ models (ranged between 0.94–0.96), which potentially indicated that the sensor performance dropped for particle size within the range of 2.5–10 μm. The poorer performance of the PMS for the coarse PM fraction was observed; the relationship between the estimated PMS and APS mass concentration values for the coarse size fraction (i.e., particle sizes from 2.5 to 10 μm) was markedly worse than smaller size fractions (S13 Fig).

**Table 4. Summary of the calibration models for mass concentration [a], $R^2$, BIC, and the NMAE of the calibration models.**

| Indices | Equation | Regression [b] | $R^2$ | BIC | NMAE |
|---|---|---|---|---|---|
| *Full concentration range (APS total number concentration between 0–1000 #/ cm$^3$) (n = 4,134)* | | | | | |
| PM$_1$ | Linear | $y = 1.06\ x$ | 0.96 | 25852 | 3.11% |
| | Polynomial | $y = 0.76\ x + 0.007\ x^2$ | 0.97 | 24480 | 2.41% |
| | Linear + CRI + density | $y = 1.13\ x + 13.88\ CRI - 10.13\ density$ | 0.97 | 24181 | 2.84% |
| | Polynomial + CRI + density | $y = 0.83\ x + 0.01x^2 + 14.44\ CRI - 9.58\ density$ | 0.98 | 23432 | 2.33% |
| PM$_{2.5}$ | Linear | $y = 2.29\ x$ | 0.94 | 42435 | 4.53% |
| | Polynomial | $y = 1.55\ x + 0.006\ x^2$ | 0.96 | 41341 | 3.41% |
| | Linear + CRI + RH | $y = 2.51\ x - 23.27\ CRI + 0.36\ RH$ | 0.95 | 41565 | 4.07% |
| | Polynomial + CRI + RH | $y = 1.80\ x + 0.004\ x2 - 15.55\ CRI + 0.42\ RH$ | 0.96 | 41152 | 3.44% |
| PM$_{10}$ | Linear | $y = 1.53\ x$ | 0.85 | 49963 | 3.56% |
| | Polynomial | $y = 0.72\ x - 0.003\ x^2$ | 0.88 | 48959 | 2.61% |
| | Linear + CRI + RH | $y = 1.69\ x - 39.14\ CRI + 0.56\ RH$ | 0.87 | 49544 | 3.31% |
| | Polynomial + CRI + RH | $y = 0.73\ x + 0.003\ x2 - 17.94\ CRI + 0.75\ RH$ | 0.88 | 48931 | 2.61% |
| *Lower concentration range (APS total number concentration $<$ 100 #/ cm$^3$) (n = 1,838)* | | | | | |
| PM$_1$ | Linear | $y = 0.72\ x$ | 0.90 | 6211 | 10.10% |
| | Polynomial | $y = 0.91\ x - 0.02\ x^2$ | 0.91 | 6053 | 9.23% |
| | Linear + CRI + density | $y = 0.57\ x + 4.80\ CRI - 2.68\ density$ | 0.93 | 5746 | 8.16% |
| | Polynomial + CRI + density | $y = 0.80\ x - 0.02\ x^2 + 4.93\ CRI - 2.95\ density$ | 0.93 | 5694 | 8.08% |
| PM$_{2.5}$ | Linear | $y = 1.10\ x$ | 0.91 | 11170 | 9.14% |
| | Polynomial | $y = 1.34\ x - 0.01\ x^2$ | 0.91 | 11087 | 8.80% |
| | Linear + CRI + RH | $y = 0.97\ x - 1.87\ CRI + 0.16\ RH$ | 0.92 | 10890 | 8.01% |
| | Polynomial + CRI + RH | $y = 1.14\ x - 0.006\ x^2 - 2.43\ CRI + 0.17\ RH$ | 0.92 | 10885 | 7.97% |
| PM$_{10}$ | Linear | $y = 0.63\ x$ | 0.89 | 11878 | 9.30% |
| | Polynomial | $y = 0.86\ x - 0.01\ x^2$ | 0.90 | 11686 | 8.53% |
| | Linear + CRI + RH | $y = 0.54\ x - 2.15\ CRI + 0.20\ RH$ | 0.91 | 11627 | 8.01% |
| | Polynomial + CRI + RH | $y = 0.78\ x - 0.004\ x2 - 3.57\ CRI + 0.21\ RH$ | 0.91 | 11461 | 7.75% |

[a] The APS mass concentration measurement was obtained with the assumed density for testing aerosols in Table 1.

[b] y: APS measurement; x: PMS measurement

Definition of abbreviations: n = number of datapoints.

Fig 5 shows a comparison between the pre-calibrated and post-calibrated PMS and APS particle number concentrations for full and lower concentration ranges. The pre-calibrated (OEM) number concentration vs. APS exhibits a linear trend over the entire range for all aerosols; however, the PMS underestimates the number of particles. The calibration significantly improves the agreement demonstrating the importance of calibration and the accuracy gains from applying calibrations. The simple linear relationship shows excellent agreement over the entire range of particle concentration and properties, the fitting parameter are shown in Table 3.

For mass concentration (Fig 6), the PMS does not increase linearly compared to APS estimates, especially at higher concentrations. We do not have a satisfactory explanation for the non-linear trend when using the OEM calibration. Also, we observed a notable discrepancy in the PMS and APS relationship between ATD and other test aerosols, which may be related to a wide range of particle CRI in ATD; see Table 1. The graphical comparison is consistent with our results from Table 4 that shows lower NMAE for the models with a quadratic term. Overall, the mass concentration models adjusted for particle and environmental specific properties such as CRI, density, RH, as well as adjustment for non-linearity, seem to be necessary.

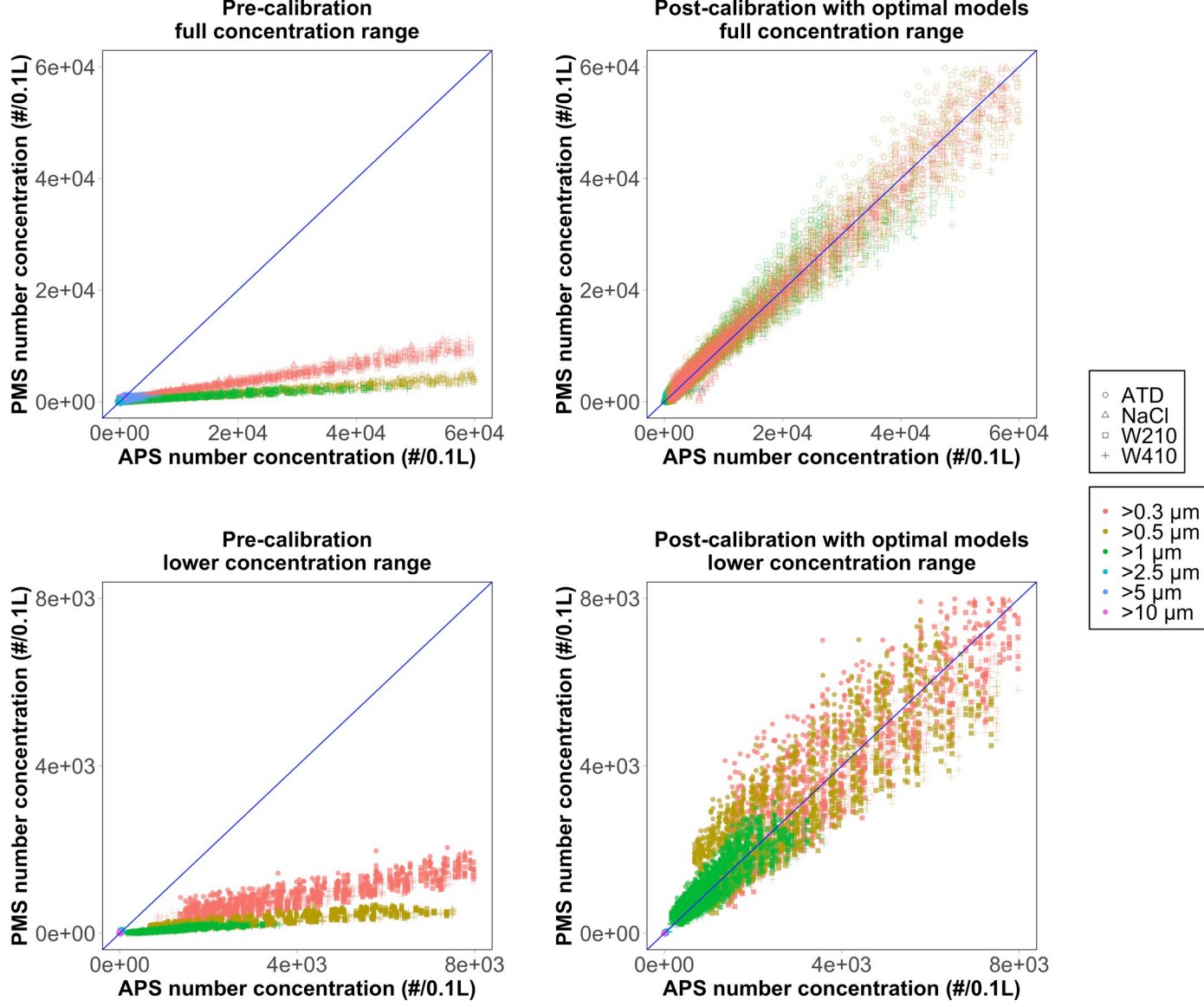

**Fig 5. A comparison of the pre-calibrated and post-calibrated number concentrations by full and lower concentration range.** The blue line represents the 1:1 relationship between the PMS and APS concentration.

## Conclusions

This study evaluated the PMS sensors and reported the calibration algorithm for both number concentration and mass concentration. We found that the PMS's number concentrations can be corrected using a simple linear model, and mass concentrations can be better corrected using a polynomial model. Although the BIC indicated models adjusted for particle properties and environmental conditions are statistically superior, those models did not significantly improve NMAE. When restricting the fit to the lower concentration, the model's accuracy decreases for both number and mass concentration, and the larger size bins tended to have higher errors. We used particles in a relatively narrow range of CRIs (1.53–1.64) and densities (1.03–2.7 g/cm$^3$), and our RH was restricted within 17–80%. If the particle properties and environmental conditions of interest are significantly different from tested scenarios, one may

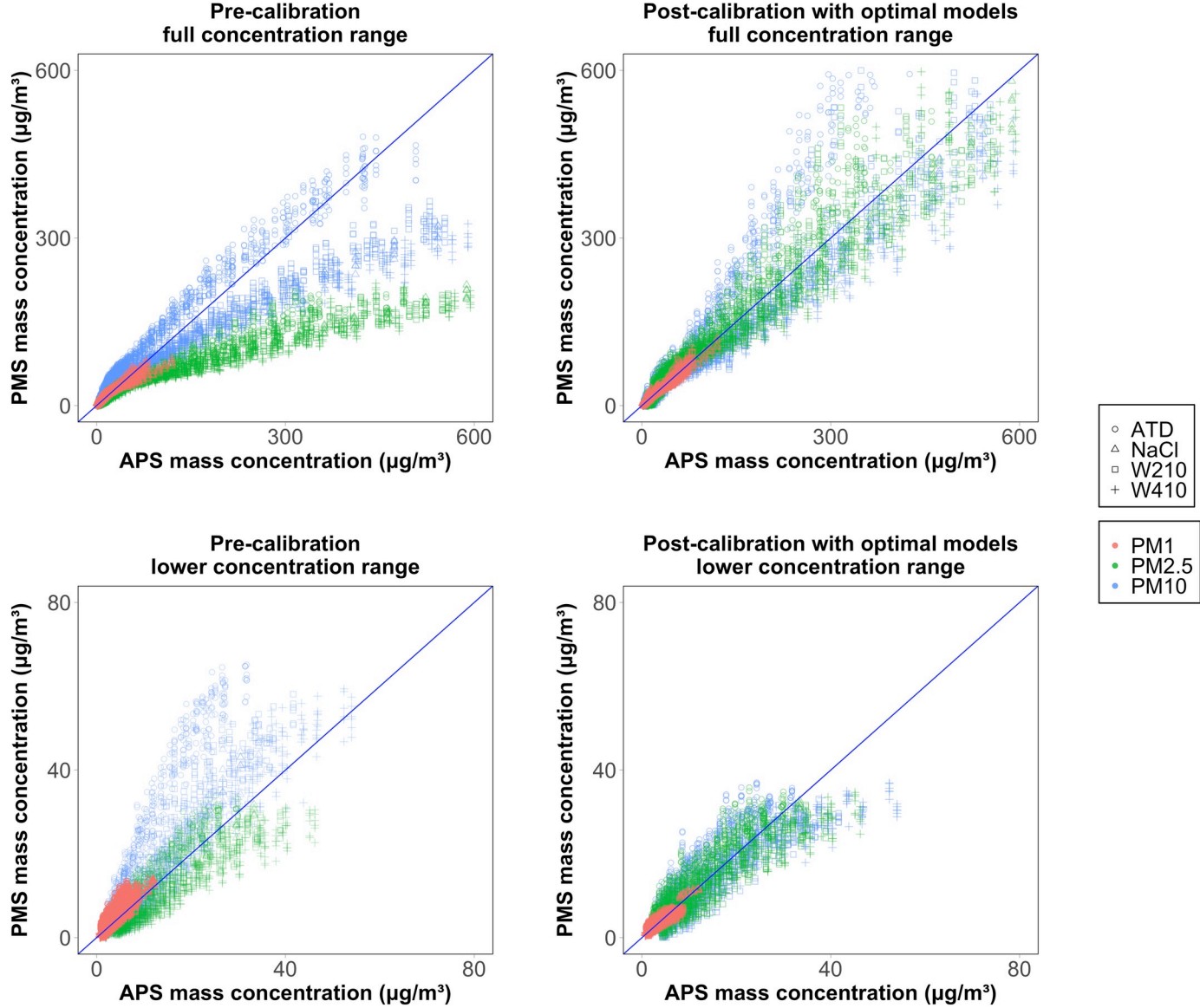

**Fig 6. A comparison of the pre-calibrated and post-calibrated mass concentrations by full and lower concentration range.** The blue line represents the 1:1 relationship between the PMS and APS concentration.

need to consider these effects. Despite these limitations, these results are relevant when size-resolved number concentration is desired, especially for using these sensors in high concentration environments, including indoor air quality monitoring, occupational/industrial exposure assessments, or near-source monitoring scenarios. Since the test aerosols used in this study are applicable for several occupational health scenarios, a better exposure assessment could be achieved. In monitoring near-source and indoor air quality, where field calibration might be challenging, the more general algorithms applicable for a broader concentration range and known particle properties could substitute for the labor-intensive gravimetric measurements. The low-cost monitors also enable the development of distributed sensor networks with a much higher special resolution than those currently available from government air quality monitoring sites.

## Supporting information

**S1 Fig. Pearson correlation between the uncalibrated PMS number concentration (6 sensors pooled together) and APS number concentration for different size ranges.**
(TIF)

**S2 Fig. Pearson correlation between the uncalibrated PMS mass concentration (6 sensors pooled together) and APS mass concentration for different size ranges.**
(TIF)

**S3 Fig. The normalized particle size distribution of the Arizona Test Dust (ATD), NaCl, W210, and W410 measured by the APS.** The median diameter of the ATD, saline, W210 and W410 aerosol are 0.94 μm, 0.86 μm, 0.92 μm and 0.96 μm, respectively.
(TIF)

**S4 Fig. Pearson correlation between pairs of PMS for number concentration (for size range > 0.3 μm).**
(TIF)

**S5 Fig. Pearson correlation between pairs of PMS for number concentration (for size range > 0.5 μm).**
(TIF)

**S6 Fig. Pearson correlation between pairs of PMS for number concentration (for size range > 1 μm).**
(TIF)

**S7 Fig. Pearson correlation between pairs of PMS for number concentration (for size range > 2.5 μm).**
(TIF)

**S8 Fig. Pearson correlation between pairs of PMS for number concentration (for size range > 5 μm).**
(TIF)

**S9 Fig. Pearson correlation between pairs of PMS for number concentration (for size range > 10 μm).**
(TIF)

**S10 Fig. Pearson correlation between pairs of PMS for mass concentration (for $PM_1$).**
(TIF)

**S11 Fig. Pearson correlation between pairs of PMS for mass concentration (for $PM_{2.5}$).**
(TIF)

**S12 Fig. Pearson correlation between pairs of PMS for mass concentration (for $PM_{10}$).**
(TIF)

**S13 Fig. A comparison of the pre-calibrated $PM_1$, $PM_{2.5}$, $PM_{10}$, and $PM_{coarse}$ size fraction (i.e., particle sizes from 2.5 to 10 μm).**
(TIF)

**S14 Fig. A comparison of the pre-calibrated and post-calibrated number concentrations by different size bins.**
(TIF)

**S1 Table. Forms of linear model fitted for number concentration and mass concentration for calibration.**
(DOCX)

**S2 Table. Summary of the R$^2$, Bayesian Information Criterion (BIC), and the Normalized Mean Absolute Error (NMAE) of the calibration models for number concentration.**
(DOCX)

**S3 Table. Summary of the R$^2$, Bayesian Information Criterion (BIC), and the Normalized Mean Absolute Error (NMAE) of the calibration models for mass concentration.**
(DOCX)

## Acknowledgments

The authors wish to express special thanks to William Lin and Koustubh Muluk, the students at the University of Washington, for helping to run the aerosol chamber tests in this study.

## Author Contributions

**Conceptualization:** Ching-Hsuan Huang, Jiayang He, Elena Austin, Edmund Seto, Igor Novosselov.

**Data curation:** Ching-Hsuan Huang.

**Formal analysis:** Ching-Hsuan Huang.

**Funding acquisition:** Igor Novosselov.

**Methodology:** Ching-Hsuan Huang, Jiayang He, Elena Austin, Edmund Seto, Igor Novosselov.

**Project administration:** Jiayang He, Igor Novosselov.

**Resources:** Jiayang He, Edmund Seto, Igor Novosselov.

**Software:** Ching-Hsuan Huang.

**Supervision:** Elena Austin, Edmund Seto, Igor Novosselov.

**Validation:** Jiayang He, Elena Austin, Edmund Seto, Igor Novosselov.

**Visualization:** Ching-Hsuan Huang.

**Writing – original draft:** Ching-Hsuan Huang.

**Writing – review & editing:** Jiayang He, Elena Austin, Edmund Seto, Igor Novosselov.

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
