## [Decision Letter · Decision Letter 0]

5 Jul 2021

PONE-D-21-17306

Assessing the Value of Complex Refractive Index and Particle Density for Calibration of Low-Cost Particle Matter Sensor for Size-Resolved Particle Count and PM2.5 Measurements

PLOS ONE

Dear Prof. Igor Novosselov

Thank you for submitting your manuscript to PLOS ONE. After careful consideration, we feel that it has merit but does not fully meet PLOS ONE’s publication criteria as it currently stands. Therefore, we invite you to submit a revised version of the manuscript that addresses the points raised during the review process.

Please carefully review the comments by both reviewers. When revising this manuscript, please take particular attention to the presentation of results and the discussions on the limitations / uncertainties.

We look forward to receiving your revised manuscript.

Kind regards,

Zongbo Shi

Academic Editor

PLOS ONE

Journal Requirements:

Additional Editor Comments (if provided):

Both reviewers found that this paper is interesting and of high value to researchers in the field. They raised a number of issues, primarily regarding the presentation and discussions on limitations.

Reviewers' comments:

Reviewer's Responses to Questions

**Comments to the Author**

1. Is the manuscript technically sound, and do the data support the conclusions?

Reviewer #1: Yes

Reviewer #2: Partly

2. Has the statistical analysis been performed appropriately and rigorously? 

Reviewer #1: Yes

Reviewer #2: Yes

3. Have the authors made all data underlying the findings in their manuscript fully available?

Reviewer #1: Yes

Reviewer #2: No

4. Is the manuscript presented in an intelligible fashion and written in standard English?

Reviewer #1: Yes

Reviewer #2: Yes

5. Review Comments to the Author

Reviewer #1: This study provides calibration algorithms for low-cost PMS A003 PM sensor as a function of particle size and concentration. The science behind the study looks fine and paper shows good interest for PLOS ONE readers. Overall, the results are promising and the manuscript is well written. However, the authors did not present detailed limitations of the method used in the study. I have some minor concerns before accepting the paper for publication.

1. Minor edits are required throughout the text to improve clarity of the text when reading and to correct grammar.

2. Aerosol Chamber Tests: what are the limitations?

3. Line 257: I think it should be equation 2.

4. Table 4 shows summary of the calibration model for mass concentration. However the table is not explicitly described in the manuscript.

5. I suggest to add a separate section for study limitations.

6. Conclusions section is well written but the authors should give a clear future direction.

7. Figures quality also needs improvement.

Reviewer #2: General Comments

This article conducted a valuable laboratory analysis of a low-cost optical particle counter against the Aerodynamic Particle Sizer research instrument. While I have no concern regarding the laboratory experiments conducted, improvement is needed regarding how the data are analyzed, presented, and interpreted to the reader. These include the following:

1) The authors appear to discount the importance that there is a mismatch between the lowest channel of the APS and the lowest channel of the PMS. Given the highest number concentrations occur in this channel (e.g., their Figure 2 shows >0.3 counts ranging ~7000-15000 #/0.1L compared to far lower values for other channels), any differences in the upper and lower size limits of the two channels is an important discrepancy to highlight and take into account throughout the article. The question here is – what is the actual lower detection limit of the APS? If, the PMS detects to a lower detection limit, then the relative differences in the dN/dlogDp size bin in Figure 3 is explainable.

2) One outstanding question regarding the PMS sensor is their ability to detect the coarse particle size range (PM2.5-10). If the PM2.5 fraction is high, the PM10 results shown may not reveal whether the sensor performs well in high dust scenarios. The number count results indicate a performance drop for larger size channels. It is recommended that authors add an analysis of the PM2.5-10 fraction to the mass concentration analyses and discussion of the sensor performance for this size range. Given Figure 2 indicates the # counts in these larger size fractions may be very low, authors should indicate their level of confidence in larger particle size fraction data for both the number and mass-based analyses.

3) Authors at the beginning and end of the manuscript state that scenarios such as wildfire smoke, near-source environments, and other high concentration environments necessitate this research. In the conclusions, they state “major implications”. However, this seems to be a disconnect given the test aerosols chosen and that no true mass measurement was used as a reference. The interpretation of the mass concentration results should be tempered and qualified that the APS has its own set of assumptions and is not a true mass measurement. In addition, the authors should describe how their test aerosols represent (or do not) some of the cited scenarios such as wildfire smoke, near-source environments (e.g., diesel emissions), and concentration ranges anticipated in those environments.

4) Data availability - the supplemental files do not contain any data sets, nor are there data sets in the body of the manuscript. This needs to be addressed to meet the journal requirements.

Specific Comments

Line 63-64: Add citation for the statement beginning “Estimates show…”

Line 69-70: Recommend revision for clarity – do authors mean the US EPA set the NAAQS under the Clean Air Act and therefore requires monitoring by states for these PM parameters?

Line 97: Recommend looking into “developing countries” is the best description. World Bank often uses “Low and Middle Income Countries” (LMICs)

Line 97-100: Revise for clarity – do you mean “field colocations…cannot be conducted”?

Paragraph beginning on line 103: Given this experiment relies upon optical measurement methodology for both the sensor and the reference, it would be helpful to separate measurement principle from implementation of the principle in a sensor package. For example, line 105 seems to discount a measurement principle of bulk light scatter, which would include nephelometers that provide accurate measurements of the particle scattering coefficient however need calibration to estimate mass concentration.

Line 164: Can authors explain the “atmospheric” concentration channel and their choice to use the “factory” option? What correction factor is applied to the atmospheric condition channels and why was this not used, given the authors’ stated motivation to inform use of these sensors in outdoor air quality measurement scenarios. How do the results for the factory channel apply to the other channel’s measured output? Given both data sets appear, this would be a valuable analysis to provide in supplemental information.

Line 188: Add NaCl to the list.

Line 314: What mass density was used for the APS mass concentration estimates? The “assumed” density on Table 1? Is the density used in Table 4’s calibration model also the values in Table 1? Last, this sentence only specifies PM2.5 but other mass fractions are also analyzed.

Line 319: Recommend incorporating into the abstract / conclusions the significant model performance decrease for larger size particles. As noted earlier, recommend PM2.5-10 mass concentration be added as an analysis which will confirm if that size fraction is poorly measured by the sensor.

Figure 6: The PM1 results are barely viewable, PM2.5 partially blocked. Perhaps reorder the parameters for better viewing.

6. PLOS authors have the option to publish the peer review history of their article (what does this mean?). If published, this will include your full peer review and any attached files.

Reviewer #1: No

Reviewer #2: No

---

## [Author Response · Author response to Decision Letter 0]

30 Jul 2021

Responses are provided in the attached file

---

## [Editor Report · Decision Letter 1]

20 Aug 2021

PONE-D-21-17306R1

Assessing the Value of Complex Refractive Index and Particle Density for Calibration of Low-Cost Particle Matter Sensor for Size-Resolved Particle Count and PM2.5 Measurements

PLOS ONE

Dear Dr. Novosselov,

Thank you for submitting your manuscript to PLOS ONE. After careful consideration, we feel that it has merit but does not fully meet PLOS ONE’s publication criteria as it currently stands. Therefore, we invite you to submit a revised version of the manuscript that addresses the points raised during the review process.

Thank you for revising the manuscript. This is much improved.

Please note that I did not send this out for re-review. Instead I reviewed the whole manuscript and the responses carefully myself. Please find my detailed comments (at the end of this letter) for you to address before I can recommend for publication in PlosOne. 

We look forward to receiving your revised manuscript.

Kind regards,

Zongbo Shi

Academic Editor

PLOS ONE

Journal Requirements:

Additional Editor Comments :

Response to Q1 by reviewer 2: I believe the reviewer is asking for the lower size cut of APS and PMS. If you look at the specification of the APS you should be able to find out the lower size limit with 50% efficiency. The lower size limit for optical detection is 370nm. In the manuscript you mentioned that size ranges from 0.5 um to 20um. It should be noted that this is not an optical size so not comparable to PPS which is optical diameter.

Figure 4 clearly demonstrate the major difference due to the size cut. Is it more meaningful to use the actual number concentration rather than percentage for this figure. The reason is that if you look at ATD, you can see that the fraction of 0.5-1um is nearly 50% for APS but less than 20% for PMS – this is because the number concentration of particles from 0.3-0.5 um is much higher than 0.37-0.52um size range from APS.

Please consider this and respond to the comment.

In Table 2 (and all the rest of the paper), are aerodynamic size used for APS all the time? If so, why? Why not using optical size from the APS?

Figures in Response to Q2 by R2: Can you please add regression equations and R2? And the top right and bottom right figures have the same x and y axis but not the same data. Why?

Are these figures in the supplementary or main text? If not, it would be useful to add them

Line 31: please clarify “shifted” to larger or smaller size. Reading the results, I do not seem to see this “shift” clearly.

Final sentence in abstract: Please be specific of what are or is the “implications”? At the moment, it is vague and readers won’t really know the implications. Is it PMS should not be used in high concentration environment? And what is considered high concentration environment?

Line 31 to 34 – If the error is small already without adjusting aerosol properties, then do we really need to care about making further adjustments? I am slightly confused with your overall message in this paper. On the one hand, you argued that accounting for CRI and density leads to an improvement in calibration; on the other hand, you argued that cautions should be given when making such adjustment. Since you also showed that no adjustment calibration is already very good, I wonder about what exactly do you want other researchers who are using PMS to do ?

Line 36-37: Can you clarify what does this mean “as the particle properties used in fitting were within a narrow range for challenge aerosols”?

Line 37-38: Should this sentence be moved before its immediate previous sentence, i.e., to line 34?

Line 216 – explain briefly how the outliers were identified? What do “outside the measurement range of APS were removed”? I am not sure you can use this as a criterion to remove outliers.

Line 283 – If the particle number is overestimated by so much (up to nearly 3000 times), then it is likely to be meaningless data, considering the huge contribution of large particles to particle mass. Can this be checked again?

Figure 3: can you use a different Y-axis range for the bottom figures, e.g., upper limit of 1000?

Line 286 – this is confusing, what does it mean when you say “reached 0 #/0.1L”?

Line 287: why this means a lower detection limit for APS? In reality the size cut of APS is 370 nm but APS is 300 nm; also your Fig. 4 shows that the number (fraction) of particles <0.5 um is much larger from PPS than in APS.

Line 329: tended to be ? those? Better?

Fig. 6 – large uncertainty in the coarse particles particularly particles > 5 um could lead to the difficulty in calibrating particle mass ? I wonder whether there is value to separate PM1, PM2.5 and PM10 in this figure (and similarly for Fig. 5)? The figures could be much smaller so that you can still show the information, as needed, but then it becomes clearer.

Reviewer 1 is concerned about the quality of the figures – e.g., figures are not very clear due to low resolution. I presume that when you submit the final version, you will submit higher resolution figures for production ?
---

## [Editor Report · Decision Letter 2]

11 Oct 2021

PONE-D-21-17306R2Assessing the Value of Complex Refractive Index and Particle Density for Calibration of Low-Cost Particle Matter Sensor for Size-Resolved Particle Count and PM2.5 MeasurementsPLOS ONE

Dear Dr.  Igor Novosselov,

Thank you for revising your manuscript to PLOS ONE. Most of the comments were well addressed and the manuscript is much improved. Please address a couple of minor issues before we can accept your manuscript.  Please see my detailed comments at the end.

We look forward to receiving your revised manuscript.

Kind regards,

Zongbo Shi

Academic Editor

PLOS ONE

Journal Requirements:

Additional Editor Comments (if provided):

This is much improved. Most of the commented are addressed well.

Line 32-38: First the results show that the NAME is about 4% without correction; then when adjusting for CRI and density, the NAME is within 5%. The following sentence then suggests that the calibration algorithms developed in this study .....

What this tells us is that the adjusting for CRI and density does not help with the mass measurements by PMS, so this is not needed? Is this correct?

Can you please make the key message of this paper clearer, in particular what calibration algorithms? Does this refer to the "linear model" as mentioned in line 33 or the model adjusted for CRI and density? If later, then what is the point?

If I understand correctly, the number size distribution measurements by PMS are not good but the mass measurements are reasonable. Is this correct?

Line 209: not a full sentence.
---

## [Editor Report · Decision Letter 3]

26 Oct 2021

Assessing the Value of Complex Refractive Index and Particle Density for Calibration of Low-Cost Particle Matter Sensor for Size-Resolved Particle Count and PM2.5 Measurements

PONE-D-21-17306R3

Dear Dr. Novosselov,

We’re pleased to inform you that your manuscript has been judged scientifically suitable for publication and will be formally accepted for publication once it meets all outstanding technical requirements.

Kind regards,

Zongbo Shi

Academic Editor

PLOS ONE
---

## [Editor Report · Acceptance letter]

2 Nov 2021

PONE-D-21-17306R3 

Assessing the Value of Complex Refractive Index and Particle Density for Calibration of Low-Cost Particle Matter Sensor for Size-Resolved Particle Count and PM2.5 Measurements 

Dear Dr. Novosselov:

I'm pleased to inform you that your manuscript has been deemed suitable for publication in PLOS ONE. Congratulations! Your manuscript is now with our production department. 

Kind regards, 

on behalf of

Dr. Zongbo Shi 

Academic Editor

PLOS ONE